# Brain Activation During Motor Preparation and Execution in Patients with Mild Cognitive Impairment: An fNIRS Study

**DOI:** 10.3390/brainsci15040333

**Published:** 2025-03-23

**Authors:** Hanfei Li, Chenyu Fan, Ke Chen, Hongyu Xie, Guohui Yang, Haozheng Li, Xiangtong Ji, Yi Wu, Meng Li

**Affiliations:** 1State Key Laboratory of Transducer Technology, Shanghai Institute of Microsystem and Information Technology, Chinese Academy of Sciences, Shanghai 200050, China; hanfeili@mail.sim.ac.cn (H.L.); ck97@mail.sim.ac.cn (K.C.); 2University of Chinese Academy of Sciences, Beijing 100083, China; 3Department of Rehabilitation Medicine, Huashan Hospital, Fudan University, Shanghai 200032, China; 19111220105@fudan.edu.cn (C.F.); xiehongyu@fudan.edu.cn (H.X.); 2221518031@sus.edu.cn (G.Y.); lihaozheng3211@163.com (H.L.); 22111220138@m.fudan.edu.cn (X.J.)

**Keywords:** mild cognitive impairment, fNIRS, motor preparation, motor execution, cortical inhibition, gait

## Abstract

**Objectives:** This study aimed to investigate how motor preparation impacted brain activation in individuals with differing cognitive statuses. **Methods:** We investigated the cortical activation pattern of 57 individuals with mild cognitive impairment (MCI) and 67 healthy controls (HCs) using functional near-infrared spectroscopy (fNIRS) during prepared walking (PW) and single walking (SW) tasks. The study focused on assessing brain activity in four regions of interest (ROIs): the prefrontal cortex (PFC), primary motor cortex, secondary motor cortex, and parietal lobe. Additionally, we examined the behavioral performance—gait speed—during the tasks, analyzed variations in cortical activation intensity, and conducted correlation analyses between Montreal Cognitive Assessment (MoCA) scores, gait speed, and oxygenation levels. **Results:** There was no significant difference in gait speed between patients with MCI and HCs. The MCI group exhibited lower activation in the primary motor cortex, secondary motor cortex, and parietal regions compared to HCs during the motor execution stage of PW (*q* < 0.05, FDR-corrected). Additionally, activation in the primary (*r* = 0.23, *p* = 0.02) and secondary motor cortices (*r* = 0.19, *p* = 0.04) during the motor execution stage of PW correlated significantly with MoCA scores. Furthermore, brain activity in the PFC (*r* = 0.22, *p* = 0.02), primary motor cortex (*r* = 0.22, *p* = 0.01), secondary motor cortex (*r* = 0.20, *p* = 0.02), and parietal lobe (*r* = 0.19, *p* = 0.03) during the motor preparation stage of gait was positively correlated with gait speed. **Conclusions:** Our results revealed that preparing for motor tasks modulated the neural activation patterns of patients with MCI and HCs without affecting their behavioral performance.

## 1. Introduction

MCI is a symptomatic condition characterized by impaired cognitive functioning originating from a variety of underlying factors. It is defined by reduced memory and other cognitive capacities, indicating an early stage of dementia progression. According to epidemiological research, 15.5% of elderly individuals in China exhibit mild cognitive impairment [1], with an estimated annual transition rate to Alzheimer’s disease or related dementias of approximately 6% [2]. The core symptoms of MCI involve impairments across cognitive domains such as memory, attention, language, executive function, and spatial skills, often accompanied by associated declines in motor function [3,4,5]. Motor dysfunction can significantly impact the daily lives of MCI patients, underscoring the importance of investigating motor function in this population.

Cognitive systems regulate and control movements, improving motor skill effectiveness and accuracy [6,7]. Individuals with MCI often experience deficits in attention and executive function, rendering them more susceptible to distractions or forgetting the objectives and requirements of exercise tasks. This can lead to a decline in their planning and organizational capacities during gait tasks, resulting in disorganized and inaccurate movements, and increasing the risk of exercise-related safety issues [8,9]. The gait performance of MCI patients is closely linked to their cognitive state [10], the risk of developing dementia [4], and the risk of falls. Currently, there is a lack of comprehensive understanding regarding the influence of exercise preparation and execution on motor performance and brain oxygen changes in patients with MCI.

In the context of sports competitions, the “Ready” verbal cue is widely employed to guide athletes into a targeted state of task-oriented preparedness. Auditory cues for movement preparation help participants get ready and activate muscle-related motor responses [11]. Studies have demonstrated that executing movements following a preparatory phase is more efficient compared to execution without prior preparation. Furthermore, changes in cortical activity can be detected before the actual performance of the action [12]. Movement preparation has been shown to enhance postural control, increase attention and the execution of movements, and improve the safety and stability of movements, ultimately leading to better performance outcomes [13]. However, the specific neural circuits underlying movement preparation remain unclear, and selectively inhibiting network activity within the motor cortex may represent a key mechanism enabling the preparation of movements [12,14]. From a neuromechanical perspective, an individual’s gait performance is predominantly regulated by subcortical pathways within the frontal cortex that are responsible for attention and executive control processes. These neural networks are closely intertwined with systems governing movement, sensory processing, and cognitive functions [15]. Correspondingly, alterations in gait characteristics observed in patients with MCI are intimately linked to structural and functional changes occurring within the brain [16]. Existing research on gait performance in patients with MCI has primarily focused on single-task gait and dual-task gait, revealing greater alterations in gait parameters when cognitive demands are added, likely due to impairments in both cognitive and motor functions [17,18]. However, the impact of movement preparation on gait and associated brain activation patterns in MCI remains unexplored. Understanding how the addition of movement preparation affects the brain in this population is an important area for further investigation.

fNIRS is a non-invasive, radiation-free neuroimaging technique that can measure cerebral blood flow. The underlying physiological mechanism is the neurovascular coupling process, which reflects brain activation patterns during task performance by detecting changes in oxygenated and deoxygenated hemoglobin [19]. The development of lightweight, portable fNIRS devices has facilitated gait assessment applications. These portable systems offer advantages such as minimal noise interference and resilience to motion artefacts, enabling the evaluation of brain function during natural gait tasks while being worn as backpacks [20,21]. Prior to the advancement of portable fNIRS technology, much of the research on exercise preparation utilized functional magnetic resonance imaging (fMRI). Notably, a study by Sahyoun et al. indicates that the PFC becomes activated during the preparatory phase of ankle flexion and extension movements [22].

Previous studies have revealed the functions of various brain regions in the gait task. The medial sensorimotor cortex and supplementary motor area do not show significant changes in blood oxygen levels with variations in gait speed; however, the PFC and primary motor cortex play a crucial role in regulating movements to accommodate increasing treadmill velocity [23]. Another study has demonstrated that the anterior frontal lobe is involved in the preparation of motor cues, while the medial motor cortex prioritizes the planning and execution of motor programs [24]. Additional research has identified the involvement of the primary motor area, supplementary motor area, and upper parietal lobe in the gait movement [22,25]. While fMRI is limited to investigating motor imagery associated with exercise preparation and cannot monitor individuals during actual walking, the development of portable fNIRS has enabled the assessment of motion readiness in natural settings. The relevant literature has reported that in young adults, the increase in oxygenated hemoglobin in the PFC is more pronounced when individuals are preparing to exercise compared to when they are not [10]. Similarly, an fNIRS study has demonstrated that the premotor cortex and posterior parietal lobe are implicated in motor preparation in stroke patients [26]. Currently, there is a gap in the research examining the neural correlates of exercise preparation in MCI patients, and fNIRS represents a suitable neuroimaging modality for this investigation.

The purpose of this study is to explore the impact of motor preparation on the behavioral and brain activation patterns of individuals with MCI and HCs. Gait research usually includes dual-task gait and single-task gait. The research on dual-task gait aimed to use sensitive paradigms to distinguish between the two subjects and reveal the brain activation patterns of different populations under different paradigms. A previous study has found that continuous subtraction and vocabulary fluency tasks can effectively detect MCI-related gait changes during dual-task [17]. Additional research has found that gait and brain functional connectivity are abnormal in cognitive impairment groups during complex cognitive tasks, and fNIRS, combined with gait analysis, provides a new method for early detection of cognitive impairment [27]. Another study has found that MCI patients exhibit reduced frontal lobe activation under walking conditions during dual-tasks, and fNIRS can detect brain dysfunction earlier than behavioral data [28]. However, our research aimed to investigate the impact of motor preparation on MCI patients, so we chose a single task gait combined with verbal preparation commands to achieve this.

Existing research suggests the PFC, motor cortex, and parietal lobe may contribute to the processes of preparing and executing gait. Consequently, we developed a portable fNIRS headset capable of monitoring these brain regions. We hypothesize that incorporating motor preparation will yield meaningful alterations in the neural activation profiles of both individuals with MCI and HCs. To validate this hypothesis, we devised two gait tasks: a PW task and an SW task. We anticipate the motor preparation instructions will modulate the blood oxygen activation patterns in cognitive and motor-related brain regions for both participant groups. Our proposed research hypotheses are: the prefrontal and parietal regions participate in the motor preparation stage; individuals with MCI exhibit heightened activation in the PFC; under conditions with a verbal preparatory cue, motor function and associated brain activity undergo corresponding changes; variations in brain activation patterns may correlate with the degree of cognitive impairment. This study represents the first investigation of multi-regional brain activation profiles in individuals with MCI during both motor preparation and execution.

## 2. Materials and Methods

### 2.1. Participants

In the pre-experiment, we calculated an effect size of 0.35 among 10 participants. Based on an expected effect size (Cohen’s d = 0.45), a significance level (α = 0.05), and a statistical power of 0.8, the power analysis revealed the sample size of 62 patients with MCI and 62 HCs. A cohort of 57 individuals diagnosed with MCI were recruited from the Xujiahui Community Health Service Center, along with 67 HCs matched for age and gender from the local community.

The MCI diagnosis was made based on the criteria defined by Petersen and colleagues [29]: (1) Cognitive decline: cognitive impairment reported by the patient or informed person, and objective evidence of cognitive impairment, and/or objective examination confirms a decline in cognitive function compared to previous levels. (2) The daily basic abilities are normal, and the complex instrumental daily abilities may be slightly impaired. (3) No dementia. Our study enrolled participants aged 60 to 85 years, all of whom had at least six years of formal education and did not exhibit substantial visual or hearing impairments. All participants did not exhibit major psychiatric disorders, significant neurological conditions, or cerebrovascular diseases. Detailed demographic characteristics, including age, gender, height, weight, Body Mass Index (BMI), educational level, and MoCA scores, are provided in Table 1. The study protocol was reviewed and approved by the Ethics Committee of Huashan Hospital, Fudan University. Before conducting the experiment, all participants were comprehensively informed about the experimental procedures and provided their written informed consent. Our clinical trial was approved by the Clinical Research Information Service of China, a publicly accessible primary registry that is part of the WHO International Clinical Trial Registry Platform. The trial was registered on 6 March 2022, with registration number ChiCTR2200057281. Identifier: https://www.chictr.org.cn/ (accessed on 30 July 2021).

### 2.2. Gait Task

This study examined participants’ gait on a 10 m walkway using two experimental conditions: SW and PW. Before the experiment, the researcher thoroughly explained the instructions to ensure participants fully understood the process. Participants wore gait-monitoring and fNIRS devices and moved freely for 3 min to alleviate any nervousness. Both gait tasks were divided into 4 blocks with a total duration of 240 s. To avoid influencing participants’ expectations, the rest periods were pseudo-random (10 s, 15 s, 20 s and 25 s).

The SW condition involved participants walking at their preferred speed upon hearing “Walk” and stopping after 30 s when prompted. This sequence was repeated 4 times. In the PW condition, participants entered a movement readiness state after hearing “Ready”, then walked at their own pace for 30 s after a 10 s delay and stopped when instructed “Stop”. This PW task was also repeated 4 times, as shown in Figure 1.

A multi-channel fNIRS neuroimaging system (DanYang HuiChuang Medical Equipment Co., Ltd., Beijing, China) was utilized to capture cortical activity by monitoring the changes in oxygenated hemoglobin signals. The sampling rate was set at 11 Hz, using 730 nm and 850 nm wavelengths. A customized fNIRS headcap was designed to meet the experimental requirements with the well-established 10/20 electrode placement protocol. A total of 40 probes (24 sources and 16 detectors) were distributed across the entire cortex, spaced 30 mm apart, forming a 52-channel network as illustrated in Figure 2. The channel positions were determined using a digitizer (Patriot, Polhemus, Colchester, VT, USA) on a standard head model and then standardized to the Montreal Neurological Institute (MNI) space. The MNI coordinates were visualized using the BrainNet Viewer toolbox within MATLAB (2022b). Subsequently, the MNI coordinates of each channel were assigned to specific brain regions based on the Broadmann Talairach template. This study utilized the GIBBON Gait Device (Dalian Qianhan Technology Co., Ltd., Dalian, China, model QH-JBE-Y).

### 2.3. Motion Artifact Correction

The data preprocessing was performed using a MATLAB HOMER2 toolbox-based pipeline [30,31]. The initial light intensity signal was transformed into an optical density (OD) signal. Subsequently, a sliding window-based motion correction algorithm was applied to each channel independently. Any 1 s OD signal window deviating from the mean by more than 5 standard deviations was identified as containing motion artifacts, and the corresponding OD signal was discarded and reconstructed using spline interpolation. A digital bandpass Butterworth filter in the 0.01 Hz to 0.1 Hz frequency range was employed to remove physiological noise sources, such as respiration, cardiac activity, and low-frequency drift.

### 2.4. Data Process

In the SW task, we used the 2 s before the initiation of the motor execution task as the baseline. In the PW task, the 2 s before the “E-PW ready” cue were selected as the baseline. We calculate the average blood oxygen level during the task phase within a block by subtracting the baseline value from the blood oxygen value during the task.

To further ensure data quality, the coefficient of variation (CV) was calculated for each 30 s block (Equation (1)), and blocks with a CV exceeding 0.1 were excluded. Additionally, if more than half of the blocks for a specific channel were excluded, the entire channel’s signal was discarded. Thereafter, the OD signal was converted into hemoglobin oxygenation concentrations via the modified Beer–Lambert law. Finally, a digital bandpass Butterworth filter in the 0.01 Hz to 0.1 Hz frequency range was employed to remove physiological noise sources, such as respiration, cardiac activity, and low-frequency drift.(1)$${CV}_{block}\left(\%\right)=\frac{\sigma_{block}}{\mu_{block}}\times 100\%$$
where $\sigma_{block}$ and $\mu_{block}$ represented the standard deviation and the mean of a 30 s block of SW and PW, respectively. If a block’s CV exceeded 0.1, the signal from that block was considered unreliable and eliminated. Furthermore, if more than half of the blocks for a specific channel were excluded, the signal for that channel was also dismissed. Additionally, if more than 10% of a subject’s channel were removed, the entire dataset for that subject was discarded.

Next, the changes in HbO concentration in each channel for each participant were assessed using the general linear model (GLM) approach. The various experimental conditions were convolved with the standard hemodynamic response function within the GLM framework to generate the corresponding regressors. Comparisons of the beta values between the two groups were performed using two-sample *t*-tests. To account for multiple comparisons, the false discovery rate was employed, with the statistical significance threshold set at *p* < 0.05.

### 2.5. Statistical Analysis

Basic demographic information, including age, height, weight, BMI, educational level, MoCA scores, and behavioral performance, were evaluated using independent sample *t*-tests. Gender differences were examined through chi-square analyses.

Independent-sample *t*-tests were conducted to examine differences in beta values within ROIs between the two participant groups. Furthermore, we computed the effect size. To reduce the occurrence of false positives, a significance level of *q* ≤ 0.05 was set, and the False Discovery Rate (FDR) correction was applied to the statistical outcomes related to the beta values in the ROIs.

To mitigate the risk of false positives, a correction method was applied to the statistical analyses of beta within ROIs.

A Pearson correlation analysis was performed to investigate the relationship between MoCA scores and beta values, as well as the associations between gait parameters and beta values.

The significance level was set at *p* < 0.05.(2)$$r=\frac{\sum_{i=1}^{n}(X_{i}-\bar{X})(Y_{i}-\bar{Y})}{\sqrt{\sum_{i=1}^{n}{\left(X_{i}-\bar{X}\right)}^{2}}\cdot \sqrt{\sum_{i=1}^{n}{\left(Y_{i}-\bar{Y}\right)}^{2}}}$$

r represents the Pearson correlation coefficient; $X_{i}$ and $Y_{i}$ represent the values of the two variables for the *i*-th data point; $\bar{X}$ and $\bar{X}$ represent the mean values of variables$\;X_{i}$ and $Y_{i}$, respectively; *n*: the total number of data points.

## 3. Results

### 3.1. Gait Parameter

The gait data generated in this study were recorded using the GIBBON Gait Device (QH-JBE-Y) and can export various gait parameters from different tasks. This study adopted walking speed as the gait parameter as previous research has found that walking speed is a sensitive indicator for distinguishing between MCI patients and healthy elderly individuals [4]. Gait speed is defined as the distance from the first step to the last step within a specific task divided by the time taken, representing the distance walked per unit of time, typically expressed in meters per second (m/s). We first examined the gait speed of the MCI and HC groups during the execution of SW and PW tasks and found no statistically significant differences between the two groups, as presented in Table 2.

### 3.2. Brain Activation

The GLM was employed to model the data, estimate the parameters, and obtain the beta value. The disparity in beta values between individuals with MCI and HCs is shown in Figure 3, particularly during the rest state and execution stage of the SW task and preparation and execution stages of the PW task.

We calculated the beta values for participants in both the MCI and HC cohorts during the motor preparation and execution phases. After acquiring the beta values, we performed a subtraction of the MCI group’s beta values from those of the HCs and graphically represented the results. This process generated schematic illustrations depicting the differences in brain activation intensity throughout the rest-state and execution stage of SW as well as the preparation and execution stage of PW, as shown in Figure 3.

We did not analyze blood–oxygen level-dependent activation during the motor preparation stage, as this stage was only present in the PW task. Instead, we compared motor execution stage activation between the two groups across the two motor tasks. The results demonstrated that during the SW motor execution stage, parietal lobe activation was greater in the HC group relative to the MCI group (*q* = 0.03, FDR-corrected), as depicted in Figure 4A. Additionally, primary motor cortex (*q* = 0.04, FDR-corrected), secondary motor cortex (*q* = 0.04, FDR-corrected), and parietal lobe activation (*q* = 0.02, FDR-corrected) were significantly higher after FDR in the HC group compared to the MCI group during the motor execution stage of PW, as shown in Figure 4B.

We calculated Cohen’s d to assess the effect sizes of brain activation between MCI and HC groups [32,33]. Cohen’s d values of 0.2, 0.5, and >0.8 denote small, medium, and large effects, respectively [34]. Lower effect sizes were observed in PFC, primary motor cortex, and secondary motor cortex of the E-SW task and PFC in the E-PW task as shown in Table 3, indicating more pronounced disparities between MCI and HC groups in the parietal lobe of the E-SW task, primary motor cortex, secondary motor cortex, and parietal lobe of the E-PW task, which aligns with the findings presented in Figure 4.

### 3.3. Correlation

A correlation analysis was performed to explore the potential of the beta for oxygenation level to distinguish between the MCI and HC groups. The metric values and MoCA scores from both groups were compiled into a single dataset. A Pearson correlation analysis was performed to investigate the relationship between these variables. This analysis yielded the Pearson product-moment correlation coefficient.

The results of the correlation analysis indicated no significant correlation between the mean HbO concentration and MoCA scores in both participant groups during the R-SW, E-SW, and P-PW (Table A1). However, a significant positive correlation was observed during the E-PW. Specifically, there was a positive correlation between the activation intensity and the MoCA scores in the primary motor cortex (*r* = 0.23, *p* = 0.02) and secondary motor cortex (*r* = 0.19, *p* = 0.04), as shown in Figure 5.

We used walking speed as a gait parameter of interest since previous research has established that walking velocity serves as a sensitive indicator for distinguishing individuals with MCI from HCs [35]. We examined the relationship between blood oxygen activation levels and gait performance by correlating beta values with walking velocity. The correlation analysis revealed that there was no correlation between blood oxygen activation intensity and walking speed, neither during the rest-state nor the execution stage of the SW task. Similarly, these two variables were uncorrelated during the motor preparation phase of the PW task, as shown in Table A2. In contrast, the execution stage of the SW task exhibited a positive correlation between blood oxygen activation and walking speed in four brain regions: the PFC, primary motor cortex, secondary motor cortex, and the parietal lobe, as shown in Figure 6. Specifically, the PFC (*r* = 0.22, *p* = 0.02), the primary motor cortex (*r* = 0.22, *p* = 0.01), the secondary motor cortex (*r* = 0.20, *p* = 0.02), and the parietal lobe (*r* = 0.19, *p* = 0.03) all displayed this positive correlation.

## 4. Discussion

In this study, we examined the brain activity patterns and behavioral changes during the stages of motor preparation and execution in MCI patients compared to HCs. Our findings revealed that, in the absence of prior motor preparation, activation within the parietal lobe decreased in the MCI group relative to the HCs. Following motor preparation, activation intensities in the MCI group were attenuated compared to the HCs during the subsequent motor execution stage, encompassing the primary motor cortex, secondary motor cortex, and parietal lobes. And we found the effect sizes were larger in these brain areas. We discovered that, in contrast to the small-sized effect sizes in other brain regions, these brain regions exhibited a medium-strength effect size. This finding was consistent with the brain regions where significant results were obtained. This further indicated that not only were there differences between MCI patients and the elderly in these brain regions, but also that the differences between the groups might be more prominent in these very brain regions [36,37,38]. We found that activation levels across all brain regions were positively correlated with gait parameters during the preparatory stage of gait. In contrast, activation in the primary and secondary motor cortices was positively correlated with cognitive level during the motor execution stage. This suggests that while overall brain activation is lower in MCI patients compared to HCs during the preparatory stage, the activations in the primary and secondary motor cortices are key factors related to cognitive abilities during the execution stage. However, there was no significant correlation between activation levels and behavioral performances in MCI patients relative to the HCs. The impact of motor preparation on brain activation profiles during movement in MCI patients remains largely unexplored. The manners in which deficits in cognitive abilities influence brain and behavioral alterations linked to motor preparation represent an area deserving of further investigation.

The primary behavioral characteristics observed in MCI individuals include decreased walking speed [18,39,40], shorter stride length [41], prolonged step time [42], and heightened gait variability [42]. Gait is recognized as a biomarker for evaluating the progression of dementia in MCI patients, with gait velocity being the most sensitive parameter [4]. Numerous studies have reported no significant differences in gait speed under single-task conditions among MCI patients, but these differences become apparent when dual-task scenarios are introduced [4,21]. Our investigation encompasses both SW and PW following rest-state and motor preparation. Our analysis reveals no significant differences in gait features between MCI and HC groups for both tasks, as shown in Table 2. Given the relatively mild cognitive deficits exhibited by MCI participants and their preserved motor function, this result aligns with expectations, suggesting that MCI individuals and HCs exhibit normal behavioral performance in gait, and that motor preparation does not impact the behavioral outcomes in single-task gait for either group.

The successful execution of gait is a multifaceted and highly coordinated process primarily driven by the motor system. Prior research has revealed that the neural activity patterns generated by the motor cortex during movement preparation and execution are dependent on the contextual factors of the movement. Following movement preparation, proficient behavioral performance necessitates the ability to flexibly adjust actions based on the surrounding environment. One such context-dependent regulatory mechanism is proactive inhibition, which represents a form of behavioral suppression employed when anticipating the need to cease or modify an action [43]. During the preparatory stage of movement, the excitability of the motor system is temporarily suppressed [42,44]. Prior investigations have attributed this suppression to the operation of two processes: one is believed to assist in resolving competition between alternative response options, while the other aims to prevent premature initiation of responses. Interestingly, research indicates that preparing to execute sequential movements may reduce short-term intracortical inhibition of the primary motor cortex [10]. Preparation for movement can enhance the performance of visual perception tasks by accelerating responses to stimulus features associated with action [45].The primary motor cortex plays a pivotal role in acquiring new motor abilities [46], while the primary function of the secondary motor cortex is to flexibly translate sensory cues and other antecedents into motor actions, enabling adaptive decision-making [47]. Cortical inhibition during movement preparation is a key neural mechanism, but its role in cognitive deficits remains unclear. In this study, ROIs encompassed the primary motor cortex and secondary motor cortex, which are brain areas associated with motor functions. Our results revealed that, in the absence of movement preparation, the activation of motor-related brain regions was comparable between the MCI and HC groups. However, during the execution stage, following movement preparation, the activation of the primary motor cortex and secondary motor cortex was lower in the MCI group compared to the HC group. These results suggest that MCI patients demonstrate enhanced inhibition within the primary motor cortex and secondary motor cortex during the movement preparation stage relative to HCs. Previous fNIRS studies have primarily focused on the PFC, overlooking the motor areas, likely due to technical constraints. However, magnetic resonance imaging (MRI) findings suggest that motor regions play a central role in movement preparation and execution [23,24,48]. Future research should consistently incorporate motor areas in its analyses. Furthermore, more detailed investigations could be conducted to elucidate the neural mechanisms underlying the enhanced motor inhibition observed in MCI patients following movement preparation.

Gait involves the integration of cognitive and motor processes, with the cognitive system orchestrating the planning and execution of movements to optimize performance [6,49]. Previous research has identified the prefrontal and parietal lobes as brain regions associated with cognition and has demonstrated the involvement of the PFC, which is linked to executive functions, in the preparatory stages of movement [50]. The parietal cortex plays a crucial role in motor control by selectively inhibiting it [51]. The initiation and execution of gait are influenced by the availability and allocation of attentional resources, and conflicts in cognitive resources can disrupt motor preparation, leading to errors in motor execution [52]. Motor inhibition is a core component of motor control, with the parietal cortex selectively inhibiting motor control. The initiation of gait may vary with available and allocated attentional resources, and conflicts in cognitive resources can disrupt motor preparation, leading to errors in motor execution. Earlier fNIRS studies focused on the PFC, showing no differences in prefrontal activation patterns between MCI patients and healthy older adults during single-task walking [28,53,54]. However, MRI-related studies suggest that the parietal lobe is also a key area involved in the preparation and execution of gait [28]. The MRI results demonstrate that, without motor preparation, the parietal cortex showed lower activation in the MCI group compared to healthy older adults. Even after engaging in motor preparation, the parietal activation during the execution stage remains lower in the MCI group relative to HCs. These findings suggest that MCI patients exhibit reduced parietal activation in both simple gait and prepared gait tasks. All participants in this study were older adults, and prior theoretical work has indicated that in older populations, there is an age-related decrease in posterior brain region activity during task performance, concurrent with an increase in frontal lobe engagement [47]. The observed reduction in parietal lobe activation without a corresponding increase in prefrontal activity is frequently associated with age-related cognitive compensation. However, in the present investigation, this pattern was not observed, potentially due to the low cognitive demands of the task, which did not necessitate extensive resource allocation. Consequently, these findings indicate that future studies should broaden the scope of fNIRS analysis to further explore the involvement of the parietal region in gait performance among individuals with mild cognitive impairment.

Our research results indicate that, under the PW, the activation of the prefrontal cor tex, primary motor cortex, secondary motor cortex, and parietal lobe during the motor preparation is positively correlated with the gait speed. This suggests that these brain regions play an important role in improving motor preparation. Specifically, as the core region of high-level cognitive functions, the prefrontal cortex may be involved in the formulation of motor plans and the regulation of attention [30,31]; the primary motor cortex and secondary motor cortex are responsible for the generation of motor plans and the preparation for execution [55,56]; the parietal lobe plays a key role in spatial perception and body positioning [57]. The enhanced coordinated activation of these brain regions enables more effective completion of motor preparation, thereby improving gait performance. This finding further supports the important role of cognitive and motor-related brain regions in the process of motor preparation and provides a new perspective for understanding the neural mechanisms of gait control.

During the execution stage of PW, the activation intensities of the primary motor cortex and secondary motor cortex are correlated with the MoCA scores, which further confirms the phenomenon of activation inhibition in patients with MCI. Compared with the normal elderly group, the activation levels of the primary motor cortex and secondary motor cortex in MCI patients are significantly lower after motor preparation, indicating that the activities of these motor-related brain regions are inhibited [57]. This inhibitory effect may reflect the impact of the decline in cognitive function in MCI patients on motor execution; that is, individuals with poor cognitive function have difficulty effectively activating the primary motor cortex and secondary motor cortex during the motor execution stage, leading to a decrease in motor control ability [58,59]. This finding validates the crucial role of the primary motor cortex and secondary motor cortex in the motor dysfunction of MCI patients and also provides an important direction for future research on the neural mechanisms of motor function impairment in MCI patients.

In our study results, an inhibition phenomenon was observed in the secondary motor cortex and secondary motor cortex regions of MCI patients. A previous study has found evidence that both the mirror neuron system (MNS) and the frontoparietal system are involved in motor tasks, and age-related cortical thinning of the MNS was observed in older adults compared to younger individuals [60]. Another study has found that, by comparing normal elderly individuals, MCI patients, and AD patients, the function of the mirror neuron system was partially impaired in MCI patients compared to the elderly [61]. Anatomically, the mirror neuron system overlaps with primary motor cortex and secondary motor cortex [62,63]. In our study, the results showed that the degree of blood oxygen activation in primary motor cortex and secondary motor cortex during the motor execution phase was correlated with MoCA scores, and there was an inhibition phenomenon in the primary motor cortex and secondary motor cortex brain regions. This may be related to the thinning of the MNS mentioned in the study by Di Tella S et al. [60]. Therefore, we speculate that the mirror neuron system is also involved in motor preparation and execution and may be the underlying physiological mechanism behind the inhibition phenomenon.

In future studies, combining EEG and fNIRS technologies can fully leverage their complementary strengths in temporal and spatial resolution. EEG is capable of capturing neural electrical activity with millisecond precision, providing high temporal resolution information for investigating the dynamic processes of movement preparation, while fNIRS can offer localized information on cortical hemodynamic changes, compensating for EEG’s limitations in spatial resolution. By simultaneously recording neural electrical activity and hemodynamic responses, we can more comprehensively uncover the neural mechanisms of movement preparation in individuals with MCI. Furthermore, during the disease process, the disease progression of MCI patients presents various possibilities. Some will gradually progress to Alzheimer’s disease (AD), some will remain in a stable state, and a small number may revert to a normal cognitive status. Therefore, longitudinal tracking also serves as an important supplement to the mechanisms investigated in the cross-sectional study. Previous studies have found in the follow-up period lasting more than one year that the gait changes of elderly people with MCI gradually worsen. The main changes are in walking speed and the number of daily steps [64]. Another long-term study on fNIRS have indicated that during the follow-up, the cortical activities in the cognitive part of the dual-task for both elderly people with MCI and normal elderly people significantly decrease. Among them, the improvement in individuals with MCI is the most remarkable. This may suggest that regular physical exercise may be particularly beneficial to the cognitive abilities and brain functions of the elderly, especially those with MCI [65]. Through such longitudinal studies, it will be helpful to identify the crucial factor of the decline in motor function of MCI patients and the corresponding neurobiological indicators. These indicators are expected to become early markers of disease progression, enabling more timely interventions for patients.

Behaviorally, there were no differences in gait parameters. This indicated that the cognitive impairment in MCI patients did not affect their behavioral performance under this task, suggesting successful compensation at the behavioral level for this task. However, we merely observed this phenomenon and identified such inhibition as a unique feature of MCI patients compared to the elderly [10], along with a distinct brain activation pattern. Considering the successful behavioral compensation, this pattern could be regarded as a characteristic of MCI. As no one had previously observed this phenomenon in MCI patients, whether this characteristic represented compensation or pathological changes requires further research and investigation. For example, previous studies have found through fMRI that the brain regions of the MNS in the elderly became thinner and the prefrontal lobe showed a higher degree of activation, which supports the compensation theory [66]. Additionally, there were also studies that had supported the dedifferentiation hypothesis through a linear mixed analysis model under different tasks [60]. More future research is needed to further explore this issue.

The study has some limitations: (1) fNIRS investigations of movement are hindered by high noise, and leveraging short-distance probes could help to reveal more details. (2) Integrating EEG could enable multimodal research, potentially supporting advancements in brain-computer interfaces and clinical rehabilitation applications. (3) This study did not control confounding variables such as autonomic dysfunction, diet, physical activity, and medications. (4) Subgroup analysis based on MCI subtypes, including amnestic and non-amnestic, was not conducted in this study. Nevertheless, exploring the neural activation patterns across these distinct MCI subtypes represents a promising direction for future research.

## 5. Conclusions

Our fNIRS study reveals significant deviations in the brain activation patterns of individuals with MCI during both prepared and unprepared gait tasks. fNIRS demonstrates promise as a valuable tool for evaluating gait function in cognitively impaired populations, supporting potential early diagnosis and intervention. Future research could explore the underlying mechanisms of cognitive dysfunction and provide supplementary clinical data to improve patients’ gait performance and quality of life.

## Figures and Tables

**Figure 1 brainsci-15-00333-f001:**
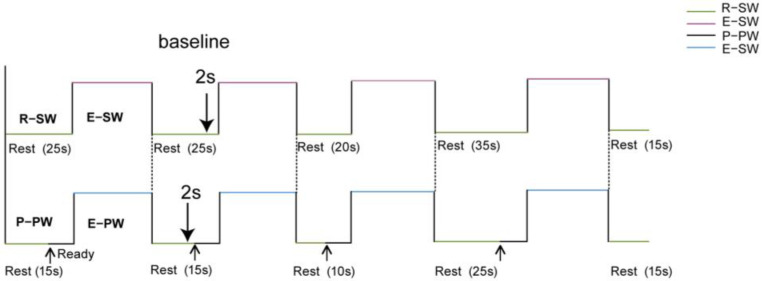
The experiment procedures. The experiment involved the SW task and PW task, lasting a total of 240 s, including 4 stages: R-SW: rest-state of SW as the control condition, E-SW: execution stage of SW, P-PW: preparation stage of PW, and E-PW: execution stage of PW. Both the SW and PW were divided into 4 segments, each comprising a random break followed by a 30 s task period.

**Figure 2 brainsci-15-00333-f002:**
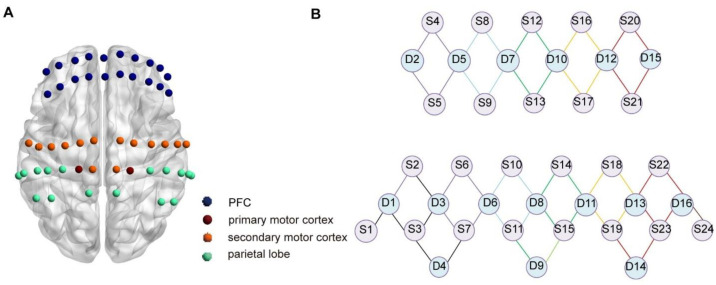
(**A**) Spatial arrangement of the 52 fNIRS channels across the cerebral cortex. (**B**) Topology of fNIRS probes, where S1–S24 represent the sources and D1–D16 denote the detectors.

**Figure 3 brainsci-15-00333-f003:**
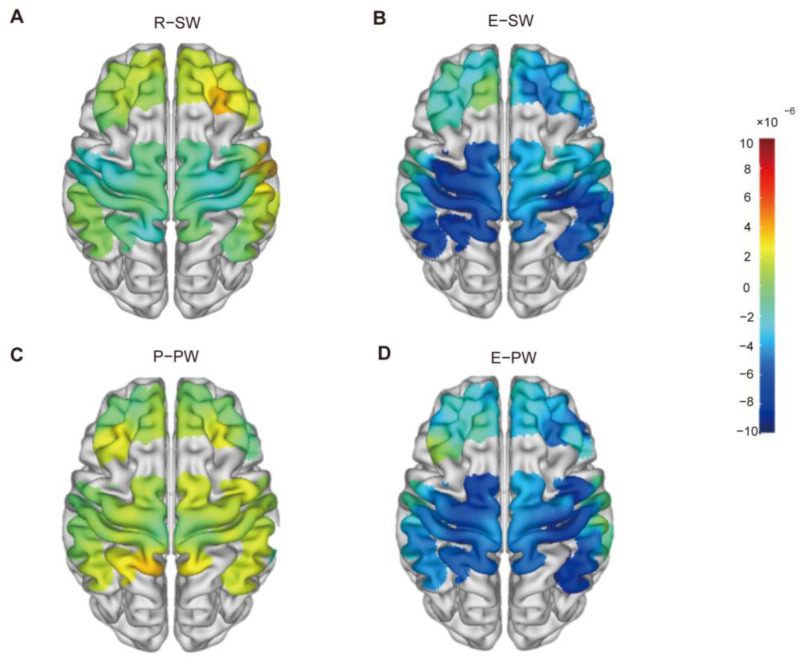
Beta values between MCI patients and the HCs in the (**A**) rest-state of SW, (**B**) execution stage of SW, (**C**) preparation stage of PW, and (**D**) execution stage of PW.

**Figure 4 brainsci-15-00333-f004:**
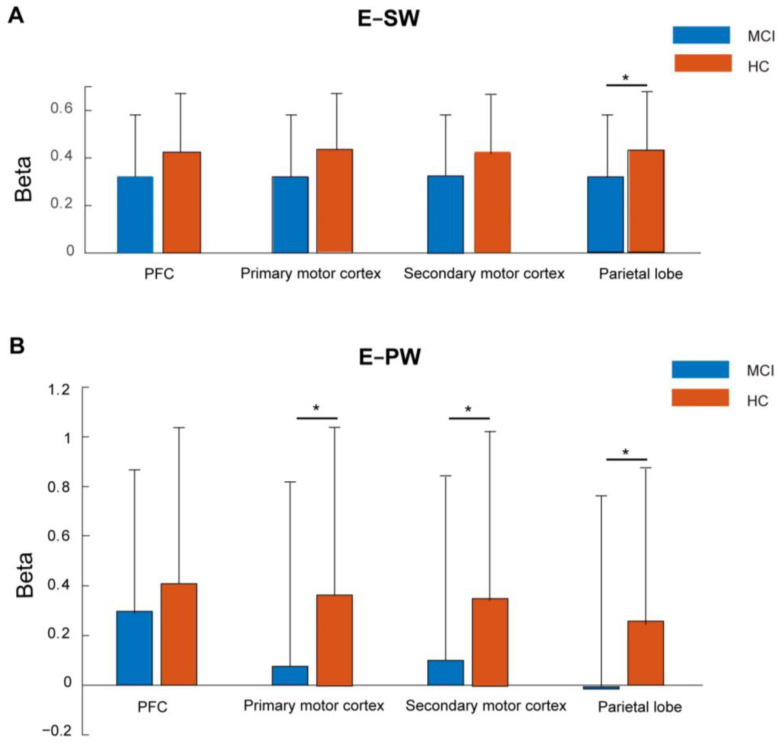
Beta values of four brain regions in the MCI group and HC group during the motor execution stage of SW (**A**) and PW (**B**). Note: * *q* < 0.05.

**Figure 5 brainsci-15-00333-f005:**
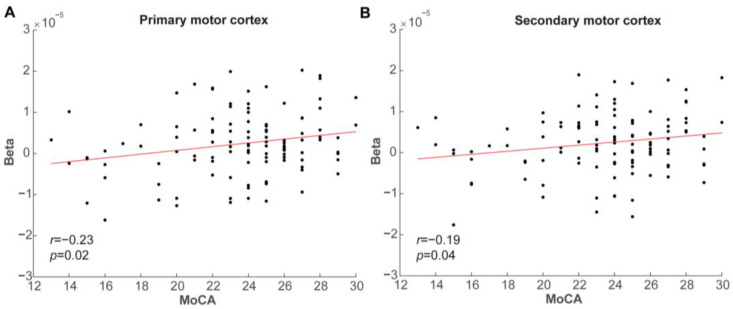
Correlation analysis between the betas for oxygenation level and MoCA scores in the primary motor cortex (**A**) and secondary motor cortex (**B**).

**Figure 6 brainsci-15-00333-f006:**
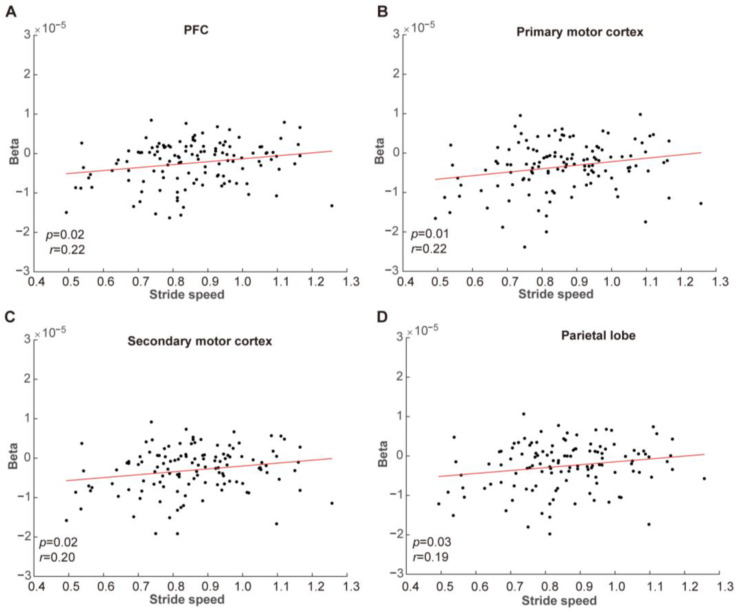
Correlation analysis between the beta for oxygenation level and walking speed in the PFC (**A**), primary motor cortex (**B**), secondary motor cortex (**C**), and parietal lobe (**D**).

**Table 1 brainsci-15-00333-t001:** Participants’ demographic information.

Characteristic	MCI (*n* = 57)	HC (*n* = 67)	*p*
Age (years)	74.55 ± 6.62	74.32 ±6.23	0.69
Gender (male/female)	25/32	29/38	0.35
Height (cm)	162.29 ± 7.86	163.13 ± 7.10	0.55
Weight (kg)	63.10 ± 9.11	63.33 ± 9.31	0.87
BMI (kg/m^2^)	22.89 ± 3.12	23.87 ± 2.91	0.72
Educational level (years)	11.28 ± 3.34	11.76 ± 2.97	0.42
MoCA	20.77 ± 3.59	26.85 ± 2.34	0.00 *

Note: * *p* < 0.05. MoCA stands for Montreal Cognitive Assessment, BMI stands for Body Mass Index.

**Table 2 brainsci-15-00333-t002:** Analysis of gait speed.

	MCI (*n* = 57)	HC (*n* = 67)	*p*
Speed (m/s)			
E-SW	0.81 ± 0.14	0.83 ± 0.23	0.16
E-PW	0.84 ± 0.26	0.85 ± 0.15	0.12

Note: E-SW represents the execution stage of single walking, and E-PW represents the execution stage of prepared walking.

**Table 3 brainsci-15-00333-t003:** Effect size in E-SW and E-PW.

	E-SW	E-PW
PFC	0.06	0.18
primary motor cortex	0.17	0.38
secondary motor cortex	0.20	0.35
parietal lobe	0.32	0.38

Note: PFC refers to the prefrontal cortex. E-SW represents the execution stage of single walking, and E-PW represents the execution stage of prepared walking.

## Data Availability

Since the data involve patient privacy, they are now stored in the unified database of Huashan Hospital. The data presented in this study are available on request from the corresponding author due to privacy.

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
