# Peer review of "Brain Activation During Motor Preparation and Execution in Patients with Mild Cognitive Impairment: An fNIRS Study"

_brainsci, 2025, doi:10.3390/brainsci15040333_

Round 1

Reviewer 1 Report

Comments and Suggestions for Authors

This study provides valuable insights into brain activation during motor preparation and execution in patients with Mild Cognitive Impairment. The use of fNIRS as a non-invasive imaging technique is well justified, and the study is well-structured. However, some areas could be further refined to strengthen the findings and interpretations.

  1. The study finds no significant differences in gait parameters between MCI and healthy controls. This raises the question of whether the tasks used were sufficiently sensitive to detect subtle motor deficits. Could incorporating a dual-task paradigm or more complex gait assessments provide additional insights?
  2. While the sample size is reasonable (57 MCI, 67 HC), a power analysis would help justify its adequacy. Given the small effect sizes in certain analyses, discussing potential limitations due to sample size would strengthen the interpretation of results.
  3. The study does not mention medication use, physical activity levels, or comorbid conditions, which could influence brain activation patterns. Adding a discussion on these potential confounders would enhance the study robustness.
  4. Since MCI is a progressive condition, a longitudinal approach would provide deeper insights into how motor preparation and execution change over time. If longitudinal data is unavailable, discussing this as a future direction would be beneficial.
  5. Since movement preparation involves both cortical and subcortical networks, combining EEG with fNIRS could provide richer temporal and spatial resolution. A brief discussion of this possibility would strengthen the future directions section.

Reviewer 2 Report

Comments and Suggestions for Authors

The manuscript titled "Brain Activation during Motor Preparation and Execution in Patients with Mild Cognitive Impairment: an fNIRS Study" presents an insightful and well-structured investigation into motor preparation and execution in individuals with MCI. This study provides valuable contributions to understanding the neural correlates of motor control in MCI patients, with potential implications for early diagnosis and intervention strategies.

Strengths

The study stands out for several reasons. Firstly, it explores a relatively under-researched topic, particularly the impact of motor preparation on brain activation in MCI patients. The use of functional near-infrared spectroscopy (fNIRS) to examine real-world gait tasks is particularly commendable, as it offers ecological validity compared to traditional neuroimaging techniques like fMRI. Furthermore, the study design is methodologically rigorous, with a reasonable sample size (57 MCI patients and 67 healthy controls) and well-defined inclusion criteria. The analysis is comprehensive, incorporating behavioral performance measures, cortical activation patterns, and correlation analyses between cognitive scores, gait parameters, and oxygenation levels. The clear presentation of results, supported by well-constructed figures and tables, also enhances the readability and impact of the findings.

Areas for Improvement

Methods Section

  1. Statistical Power Analysis: The manuscript does not provide a power analysis to justify the sample size. Including such an analysis would bolster the methodological rigor and confirm whether the sample is sufficient to detect meaningful effects.
  2. Consideration of Confounding Factors: Some variables, such as medication use, physical activity levels, or comorbidities, may influence gait and brain activity. Addressing these factors in the discussion could improve the interpretation of the results.
  3. Subgroup Analysis: Investigating whether different types of MCI (e.g., amnestic vs. non-amnestic) exhibit distinct activation patterns might yield valuable insights.
  4. Control Condition for Auditory Cueing: The study assumes that the "Ready" cue enhances motor preparation, but a control condition without verbal cues would help establish a clearer causal relationship.

Results Section

  1. Inconsistent Correlations: The correlation between gait performance and neural activity appears inconsistent across conditions, with significant relationships only emerging in specific contexts. A deeper discussion of possible reasons for these inconsistencies would be beneficial.
  2. Effect Size Reporting: Some statistical descriptions lack effect sizes. Including these values would provide additional clarity on the strength of the reported effects.

Discussion Section

  1. The discussion suggests that MCI patients exhibit enhanced inhibition in the primary and secondary motor cortices during movement preparation. However, it would be helpful to clarify whether this inhibition represents a compensatory mechanism or a pathological alteration.
  2. Relating the findings to previous research on dual-task gait in MCI patients could provide additional context.
  3. Although the study is cross-sectional, a discussion on how these findings could inform future longitudinal research on MCI progression would be valuable.
  4. One aspect that could enrich the discussion is the potential involvement of the mirror neuron system in motor preparation and execution in MCI patients. Previous research has suggested that aging and cognitive decline can impact the functionality of mirror neurons, leading to compensatory activation in the prefrontal cortex. Including this perspective could help contextualize the findings and link them to broader theories of motor control and cognitive aging. Relevant studies to consider: Di Tella S, Blasi V, Cabinio M, Bergsland N, Buccino G, Baglio F. How Do We Motorically Resonate in Aging? A Compensatory Role of Prefrontal Cortex. Front Aging Neurosci. 2021;13:694676. doi:10.3389/fnagi.2021.694676. Incorporating these references would provide a more comprehensive understanding of the interplay between motor preparation, cortical inhibition, and cognitive impairment.

Minor comments:

A few minor language refinements and typographical corrections would enhance the clarity and readability of the manuscript:

  • In the Abstract, the phrase "signifying the prodromal phase of dementia" could be reworded to "indicating an early stage of dementia progression" for improved clarity.
  • The Methods section lacks details on motion artifact correction. Expanding on the preprocessing steps beyond standard spline interpolation would improve transparency.
  • Several minor typographical errors and grammatical inconsistencies are present throughout the text. For instance, in one sentence, "The vocalized form of the movement preparation command enables participants to ready themselves for physical exertion before undertaking the actual activity, and dependable auditory stimuli can activate motor representations in a manner anticipated by the musculature" could be simplified for improved readability.

Reviewer 3 Report

Comments and Suggestions for Authors

The authors used fNIRS during prepared walking and since walking tasks in individuals with mild cognitive impairment (MCI) to examine how motor preparation impacted brain activation in the PFC, PMC, SMC, and PL. The MCI group was found to have lower activation in the PMC, SMC, and parietal regions during the motor execution stage of the PW, compared to healthy controls. Further, activation of the PMC and SMC during motor execution stage of the PW was associated with MoCA scores. While the findings are of interest, a few questions remain.

Abstract.

The specific gait parameters associated with brain activity should be clarified in the abstracts as well as the strength of correlations observed.
The conclusion needs to be clarified as no report of behavioral performance change was reported in the abstract.

Introduction.

3. How does the current work compare and contrast with recent work using fNIRS in adults with MCI? [1-3]

4. What is the rationale for the proposed hypotheses? When suggesting alterations are expected, what is the directionality of expected changes in neural activation and performance based on the literature in MCI and other neurological populations? [1-5]

Materials and Methods

5. What was the source-detector separation distance? Was this standardized?

6. Was there any monitoring of superficial scalp hemodynamics using shorter source-detector pairs to aid in further artifact removal in the fNIRS signal?

7. Were the data examined for either dark noise levels or saturation before further pre-processing? If not, what is the impact on interpretation?

8. What was the baseline used for calculation of HbO concentration levels? Was there any instruction provided to participants during this time?  

9. What were the gait parameters extracted for analysis? How were they calculated? 

Results.

10. Be sure to define all abbreviations in Tables, using the table notes.

11. How did participant group impact correlations observed between neural activation and MoCA or stride speed?

Discussion.

12. How do the present findings compare and contrast with prior work in MCI and other neurological populations [1-5]?

13. How did the selection of baseline potentially impact findings? 

Literature cited

[1] Holtzer, Roee, and Meltem Izzetoglu. "Mild cognitive impairments attenuate prefrontal cortex activations during walking in older adults." Brain sciences 10.7 (2020): 415.

[2] Wang, Zehua, et al. "Assessment of brain function in patients with cognitive impairment based on fNIRS and gait analysis." Frontiers in Aging Neuroscience 14 (2022): 799732.

[3] Bishnoi, A., & Hernandez, M. E. (2020). Dual task walking costs in older adults with mild cognitive impairment: a systematic review and meta-analysis. Aging & Mental Health, 1-12. https://doi.org/10.1080/13607863.2020.1802576

Round 2

Reviewer 2 Report

Comments and Suggestions for Authors

The manuscript has significantly improved, and many of the previous concerns have been adequately addressed. However, I have a few final suggestions that could further enhance the clarity and impact of the study before publication:

  1. Effect Size Interpretation: While you have added effect size values (Cohen’s d), consider emphasizing their implications in the discussion section. This would help strengthen the interpretation of your results and provide a clearer understanding of their magnitude.
  2. Interpretation of Cortical Inhibition: You have introduced a discussion on whether the observed inhibition in the primary and secondary motor cortices represents a compensatory mechanism or a pathological alteration. However, this remains an open question. It would be beneficial to briefly suggest how future studies could investigate this issue in more depth.
  3. Connection to Longitudinal Studies: The revised manuscript acknowledges the importance of longitudinal research, but it would be valuable to specify which key measures (e.g., gait parameters, fNIRS biomarkers) would be most informative for tracking MCI progression over time. This addition could strengthen the study’s impact.
  4. Statistical Adjustments for Multiple Comparisons: Ensure that all statistical analyses, particularly those involving multiple comparisons, have been appropriately corrected and that p-values are reported consistently throughout the manuscript.

By addressing these points, your manuscript will be even more robust and ready for publication.

Reviewer 3 Report

Comments and Suggestions for Authors

The authors are commended for their revisions and response to feedback. 

Author Response

Thank you for your insightful suggestion and helpful feedback.